# Sensitivity of *Vanessa cardui* to Temperature Variations: A Cost-Effective Experiment for Environmental Education

**DOI:** 10.3390/insects15040221

**Published:** 2024-03-25

**Authors:** Carmella Granato, Marco Campera, Matthew Bulbert

**Affiliations:** Department of Biological and Medical Sciences, Oxford Brookes University, Oxford OX3 0BP, UK; carmellasoffia@hotmail.com (C.G.); mbulbert@brookes.ac.uk (M.B.)

**Keywords:** active learning, development, phenology, pollinator, climate change, incubator, survival rate, pupation, butterfly

## Abstract

**Simple Summary:**

Temperature increases driven by climate change threaten species survival. The concept that temperatures may exceed the physiological limits of species, leading to death, is straightforward to grasp. More nuanced impacts can come from shifts in morphology and or timing of activity that may lead to slower declines. Such shifts can lead to mismatches between peak activity times of animals and their resources, referred to as phenological shifts (e.g., pollinators and blooming of flowers). The link between temperature increases and the decline of species through phenological shifts can be a challenging concept to grasp. A barrier for educators wishing to demonstrate this concept is the expense of incubators. Here we demonstrated the use of a cost-effective homemade incubator set up to investigate the impact of temperature on the development rate and morphology of the painted lady butterfly *Vanessa cardui*. Using the set-up, we found that the survival rate, development rate, body size and appearance were influenced by temperature in a predictable manner. This study provides the means and a blueprint by which educators can inform students on the impacts of temperature through an experiential approach.

**Abstract:**

Temperature increases mediated through climate change threaten the survival of species. It is of foremost importance to engage citizens and future generations in understanding the mechanisms through which temperatures impose their effects. For educators, this is not straightforward, as tools for examining the impact of temperature over the lifetime of an animal are prohibitively expensive. At the same time, environmental educators need guidance on the appropriate study systems to use with a balance between the species having an obvious response and ensuring the outcomes are ethical and sustainable. In our study, we created and tested a cost-effective experiment meant to be used for environmental education purposes. More specifically, we tested the sensitivity of the painted lady butterfly *Vanessa cardui* to temperature variations using a homemade incubator. We describe the design of this experiment and report findings on survival rate, morphological variations, development time of various stages and wingspan of adults across a range of biologically relevant temperatures. The information provided gives educators options for testing a variety of hypotheses with regards to the impacts of temperature using an affordable and flexible set-up. Furthermore, the findings can be used by students to develop an understanding of the ramifications of the butterflies’ responses in an ecological context.

## 1. Introduction

The active learning paradigm of constructing individual meaning and building internal and personal representations of knowledge requires personal experience [1]. Active or experimental learning, in which an individual’s own experiences with participating in practical activities that affirm their learning, strongly influence how individuals frame information and develop attitudes [2]. Like that of the kinaesthetic learning (i.e., associating physical movement with learning goals), experimental learning is more effective at promoting investment in actioning solutions to problems than simply visual and auditory learning styles among young students [3]. Active-learning teaching styles produce students with higher levels of emotional intelligence, i.e., self-awareness, self-regulation, motivation, empathy and social skills [4]. Highlighting the likelihood of an individual choosing to invest in future pro-environmental behaviours as an adult is framed by their own personal experience [5]. 

A challenging but crucial climate change concept for environmental educators is to develop active learning approaches that acutely illustrate the insidious impacts that rising temperatures can have on the biology of organisms [6]. Science recognises that surpassing certain temperature thresholds can lead to species extinctions [7,8]. For the wider populace, however, the link between what seems little increases in mean temperature and the widescale decline and collapse of species is largely intangible. Much of this is possibly because the influence of temperature and the mechanism by which it acts are nuanced, subtle and often realised across subsequent generations rather than having instantaneous consequences [9]. Temperature increases can act directly in changing physical and sensory features that can adversely alter mobility and sensitivity to cues. Conversely, these modifications can also be adaptive (i.e., phenotypically plastic), with expressed traits acting to buffer against, avoid or reduce temperature related stress [10,11,12]. Temperature increases of the magnitude we are currently experiencing are leading to large-scale shifts in phenology, i.e., seasonal timings [9,13]. Such shifts have been shown to result in a mismatch between an organism’s readiness to use a resource and the availability of the resource. Indeed, examples of key relationships have been shown to be out of sync such as pollinators and floral resources [14,15], migratory birds and their prey [16] and herbivores and their host plants [17,18]. The importance of understanding the impact of phenology is evidenced by global initiatives that have been set up to track these shifts in nature, many of which have relied on citizens to collect data [19,20,21]. Such initiatives even suggest that establishing phenology networks has been key to elevating public awareness of the impacts of climate change. The part of the story missing for participants of such initiatives is the mechanism through which temperature imposes these phenological shifts. 

There are significant barriers for educators wishing to explore the mechanisms by which temperature influences species survival and trait development. The first is that controlled experiments typically require expensive infrastructure such as temperature-controlled incubators. Incubators designed for rearing insects used by scientists range from GBP 500 to well over GBP 2000 for equivalent small units. The second barrier is knowledge of a study system that provides reliable outcomes when exposed to a range of temperatures. Phenology shifts are well documented in butterflies, with phenology traits promoted as indicators of species’ responses to the changing environment [22]. As poikilotherms, temperature highly influences their life cycle [23]. Over the past two decades, first appearance, mean flight date and flight periods have increased, in parallel with an increase of 1–1.5 °C in central-England spring and summer temperatures [23]. Migratory species can be acutely vulnerable to environmental change since they depend on a series of habitats throughout their migration [24]. Thus, it is likely that they may also have plastic responses to temperature rather than just providing survival data.

For this study, we explored the use of *Vanessa cardui* (painted lady) as a candidate to study thermal tolerance and plasticity, as it appeared to meet our pre-determined eligibility criteria. Our criteria for selection are that the species must have the following characteristics: A conservation status of Least Concern: the British Isles are wholly dependent on immigrants from the continent, and they arrive in substantial numbers [24,25].A wide distribution range to increase the prospect that the species may require adaptations to deal with different environmental gradients and to increase the chance the study can be replicated globally: *V. cardui* is a long-distance migrant with a wide distribution range, inhabiting all continents except for Antarctica and the majority of South America [25].Larvae are easily attainable in high numbers and ethically sourced, and breeding requires minimal demands for teachers and students, with the animal being relatively robust. Retailers that breed butterflies to sell as pets are found almost anywhere, from coastal to urban areas, and *V. cardui* is one of the few species that can breed intensively in a variety of habitats.There is some indication that the species show gradual and measurable changes across the thermal range for which our study was to be conducted: adults are first seen in late March and numbers continue to rise through May and June as further migrants arrive from the continent [26]. A few publications have shown that *V. cardui* vary in developmental timing and wing morphology across different temperatures (e.g., [27,28]).It is cosmopolitan and occurs naturally in large numbers.

This study had two main aims. The first was to develop and test a cost-effective and experimental flexible set-up that educators can use to give students first-hand experience of the mechanisms by which shifts in temperature can alter the fitness of organisms. The second was to demonstrate the effectiveness of the set-up under proper experimental conditions to address the question of how temperature influences the biology of *V. cardui*. It was hypothesised that if the set-up was effective subjecting the larvae to increased temperatures should lead to predictable phenological adaptations. If true, we predicted that: 1. Larvae of *V. cardui* incubated in warm temperatures will experience shorter development times and consequently early emergence; 2. Larvae incubated at high temperatures will have a higher mortality rate; and 3. With warming temperatures, there will be an increased likelihood of phenotypic shifts and trait aberrations. We also hoped that outcomes of the study could help educators design their own experiments with the knowledge of the sensitivity of *V. cardui* exposed to a range of temperatures and an understanding of the optimal temperatures for studying survival estimates versus phenotypic responses where optimal is defined as comprise between survivability (i.e., low mortality rate) and seeing clear phenological responses in a timely manner.

## 2. Materials and Methods

### 2.1. Source and Husbandry of V. cardui

For this study, three Butterfly Garden Kits were purchased from a conservation education company Insect Lore, Indian Queens, Cornwall, UK (GBP 64.99 each). Each kit contains: pop-up, reusable 63 cm tall clear mesh habitat; 35 live *V. cardui* larvae; specially formulated, ready to use, larvae food; 35 vials with lids; 70 sticking pads; feeding pipette; plastic spoon; food levelling tool; small transfer brush. This provided third-instar *V. cardui* larvae. Each larva was kept in its own housing, a plastic container of 5 cm × 5 cm × 6 cm, with airholes in the lid, and an allocation of larvae food. The larvae food was provided by the company and included a classified formular specialised for *V. cardui* larvae. Each container received one teaspoon of this food, which was enough to sustain the larvae until pupation i.e., no food replacement was required (Figure 1).

### 2.2. Incubator Design

To ensure the study could be replicated by schools, it was necessary to design a cost-effective and robust incubator set-up. Considerations of the design needed to include affordability of the incubator design and its capacity to ensure temperatures were reliably maintained. Here we outline the design and some of the design considerations that were made (Table 1).

The incubator was a modular design made with off-the-shelf items and easily constructed and included the following design elements (image in Table 1). 1. Basking clip-on spot lamp, used for ease of assembly. The basking spotlight selected was due to the shape of the bulb, categorised as a reflector bulb (code R); this bulb generates diffused heat in all directions, creating an evenly distributed heat source throughout the incubator. The wattage of the bulbs was selected based on the desired temperature output for each treatment: 25–35 °C = 50 W and 40 °C = 100 W. Ideally using the same wattage would be simpler, but it was not possible to create a 40 °C environment using the 50 W bulb. 2. The HabiStat (Swell UK, Hyde) dimming thermostat was selected for its low–high monitoring range and accuracy. The thermostat was responsible for keeping temperatures consistent within 0.9 °C for all temperature treatments, for example 25.0–25.9 °C. The thermostat automatically turned off the light briefly when temperatures exceeded this threshold. However, the light was consistently on for 24 h, and the thermostat was only triggered when the incubator was manipulated; otherwise it remained stable when left alone. 3. The incubator body was created from polystyrene foam boards, which were selected for their high thermal resistance, ease of assembly and accessibility. Each incubator was 42 cm × 30 cm × 44 cm in size with the door cut to half size. 4. The 30 cm butterfly habitat nets were selected to fit within the incubators, allowing for heat, airflow and moisture exchange. 5. The digital thermometers were selected for accuracy of interpretating a variety of temperature ranges; they were connected to the inside of the mesh butterfly net for accuracy, but displayed on the outer body for ease of interpreting. 6. A half sheet of polystyrene foam was used to cover half of the incubator’s front. For this aspect of the design, we experimented with a full-length polystyrene cover and no cover at all. The full cover decreased airflow and increased the temperature past the desired outcome. In contrast, with no cover, we were unable to achieve the desired temperature. The half-door design provided adequate airflow whilst maintaining the desired temperature for each treatment. Incubators were set up three days prior to receiving live specimens, which enabled necessary adjustments to heat source placement and thermostat settings to ensure a consistent temperature was present and evenly distributed throughout each incubator. Temperatures continued to be monitored daily throughout the entirety of the experiment. An additional temperature gun was used to ensure accuracy. Finally, all electrical elements were checked, and Portable Appliance Testing (PAT) tested by an electrical and maintenance company (Vinchi, Hemel Hempstead, UK) for compliance to ensure the set-up was safe. 

#### 2.2.1. Experimental Design

Fourth-instar larvae (*n* = 20) were exposed to one of five temperature ranges. The larvae were received as third instars. Previous studies indicated a higher mortality rate for *V. cardui* larvae at temperatures exceeding 28 °C prior to the fourth-instar stage [29]. All third-instar *V. cardui* larvae were kept at room temperature (18.5 °C) until reaching the fourth instar. All individuals were re-measured three days post arrival, with all individuals measuring between 13–16 mm, confirming the fourth-instar stage (Table A1 in the Appendix A). Temperature gradients included room temperature (18.5 °C), 25 °C, 30 °C, 35 °C and 40 °C. The room temperature enclosure consisted of one 63 cm mesh habitat and a White Python Digital Thermometer Hygrometer with no incubator housing. *V. cardui* larvae were randomly assigned to temperature treatments using a random number generator. Individuals were placed into the centre of their allocated incubators. Room temperature was selected as a treatment with the idea that it might provide a treatment that the schools do not need an incubator set-up for; there is, of course, a limitation here, as it assumes the room temperature is similar wherever the study is done. 

#### 2.2.2. Measurements

Morphological changes in larvae were recorded three days post incubation (Table A2). A colour code key was referenced when noting changes in *V. cardui* larvae (Figure 2). Survival rate (individuals alive per day) and phenological changes such as time to pupation, pupation duration and emergence rate, of *V. cardui* larvae were recorded (Table A3). Post emergence morphological attributes of *V. cardui* adults were recorded, including wingspan (mm) and any visible abnormalities (i.e., deformed wing, missing limb) (Table A4).

#### 2.2.3. Data Analysis for Baseline Butterfly Experiment

Analyses were conducted in R Studio, version 4.1 (R Studio Team, Boston, USA, 2020). Calculations of pupation duration (days) were made, and wingspans (mm) were recorded. Individuals were coded to represent colour (i.e., B = black, B&W = black and white, W = white; Figure 2). A survival analysis was used to plot the probability of survival of *V. cardui* larvae to emergence using ‘Survminer’ [30] and ‘ggplot2’ [31]. The analysis was parametrised with survival and event being the number of days since the fourth instar. Generalised linear mixed models were used for condition of *V. cardui* (condition classified as deformed or normal), pupation duration and wingspan size. Package instalments included ‘Generalized Linear Mixed Model using Template Model Builder’ (‘glmmTMB’) [32], ‘Diagnostics for Hierarchical Regression Models’ (‘DHARMA’) [33] and ‘Estimated Marginal Means’ (‘emmeans’) [34]. To ensure the best fit, a variety of family models were tested for each response variable. Selected models for each response variable were Generalized Poisson distribution (genpois) = pupation duration and genpois = wingspan [32]. Colour variation of *V. cardui* larvae in relation to temperature treatments was displayed in a bar chart.

## 3. Results

### 3.1. Survival Rate of V. cardui at Different Temperatures 

Probability of survival of *V. cardui* larvae to emergence and the rate of development varied among the different temperature treatments (Figure 3). The highest temperature treatment of 40 °C displayed a 100% mortality rate, with 98% of those mortalities occurring before pupation. Both the 35 °C and 30 °C treatments completed their full life cycle in the same amount of time, although the 30 °C temperature treatment displayed far less attrition. Larvae at room temperature displayed a 100% survival rate but took twice as long to complete their cycles relative to the 30–35 °C temperature treatments. Interestingly, the 25 °C treatment led to relatively minor attrition but took around 10 days less to complete the cycle relative to the room temperature treatment.

### 3.2. Morphological Variations of V. cardui Larvae in Response to Temperature Variation

Morphological adaptations were present in *V. cardui* larvae between the fourth and fifth instar under different temperature treatments. Prior to the fourth-instar stage and exposure to any temperature treatments, all 100 individuals were black in colour. Changes in colour varied among the treatments; larvae held at room temperature (RT) predominantly stayed the same prior to treatment exposure, with 20% of larvae changing to black and white variations. As temperature increased, the number of larvae to develop white colour variations increased as did the distribution of white variation patterns until larvae were completely white in colour (Figure 4).

### 3.3. Morphological and Phenological Variations of V. cardui in Relation to Temperature

With RT as the exception, deformity was present in all surviving *V. cardui* in all temperature treatments. The rate of expression of deformity increased with temperature increase (in the following, the value n represents the total number of individuals that made it to adulthood, the percentage represents the deformity rate out of the total number) RT (*n* = 20) 0%, 25 °C (*n* = 18) 11.1%, 30 °C (*n* = 17) 29.4%, 35 °C (*n* = 7) 57.1%. Wingspan of fully formed *V. cardui* also varied amongst temperature treatments, except for one individual in the RT treatment, which displayed a wingspan of 60 mm. The RT treatment displayed the largest collective of individuals with the smallest wingspan of 62 mm (*n* = 20, 50%), in contrast to 25 °C and 30 °C (*n* = 16, 12.5%; *n* = 12, 0%). Wingspan displays greater than 62 mm were as followis: RT, *n* = 20, 50%; 25 °C, *n* = 16, 87.5%; 30 °C, *n* = 12, 100%; and 35 °C, *n* = 4, 75%. Temperature treatment 30 °C displayed the largest wingspan of 70 mm (16.6%). Phenological variations in the form of pupation duration (days) also varied in relation to temperature treatments. Larvae held at RT displayed the longest pupation period, with a mean pupation duration of 12.1 days; in contrast, 25 °C = 7.3 days, 30 °C = 5.7 days, 35 °C = 5.5 days. The rate of pupation duration decreased with temperature increase (Figure 5, Table 2).

## 4. Discussion

### 4.1. Temperature Effects on Phenological and Morphological Attributes of V. cardui 

Here we showed that a basic ecological experiment manipulating temperature variations can alter the phenological traits of *V. cardui,* as seen in other butterfly species. The optimal temperature (optimal defined as a comprise between survivability, i.e., low mortality rate and seeing clear phenological responses in a timely manner) was consistent with studies using more sophisticated and expensive equipment at around 28 °C [29,35,36]. As anticipated, larvae that were exposed to higher temperatures had an increase in development rate, and lower pupation period in line with other similar studies [37]. Likewise, a temperature threshold was reached in which the ability of the larvae to complete their maturation was clearly compromised while wing deformities were more apparent with an increase in temperature. 

Temperature variation also influenced phenotypic traits. Interestingly, our wingspan results were contrary to the ‘temperature–size rule’ [38], which suggest that adults of ectotherms raised at higher temperatures are smaller than their counterparts raised at cooler temperatures [37,38]. This rule is far from absolute, with several studies showing contradictory findings [39,40]. Indeed, the likelihood of temperature having a positive, negative or neutral impact on wingspan can be species-specific [39] or dependent on the developmental stage at which the temperature increase occurs [40]. For instance, Wilson et al. [40] found that adults from different families of butterflies that were exposed to higher temperatures at late larval stages, as per our experiment, had substantially larger wingspans than adults exposed at earlier larval or pupal stages. In fact, it seems that different evolutionary outcomes maybe expected among populations or species with different migratory tendencies. In some ways increased wingspan for migratory species at higher temperatures makes sense given they are likely to invest in traits that facilitate migration [41,42]. However, wingspan measurements are rudimentary representatives of mobility potential. Indeed, mobility can be affected by temperature in other ways, such as by altering wing shape [43] or flight endurance [44]. 

Larvae also varied with the degree of melanisation decreasing with an increase in temperature. Changes in colouration have been documented in adults, with lighter wing colouration occurring in individuals in warmer conditions [45]. As far as we are aware, this is the first documentation of colour shifts in larvae. Although not previously described for *V. cardui*, this phenomenon has been observed in other species such as monarchs within both laboratory [46,47] and field conditions [12] and is suggested to be a direct consequence of melanisation being linked to thermoregulation, with greater melanisation needed in colder conditions.

### 4.2. Flexibility of Experimental Set-Up and Caveats

Here we demonstrated that the combination of the homemade set-up with the chosen study system is a powerful and scalable tool for testing the impacts of temperature on developmental timing and morphology. The cost effectiveness of the set-up means educators can potentially afford more than one incubator to examine impacts across gradients of temperature. The experiment also provides a range of data types making it scalable from early to late school students and even into early university. For example, the data collection can be as simple as calculating the time it takes from larvae to adult emergence and in this case, there is only a need to compare two temperatures with one possibly being just room temperature. This experiment is ideal for young students (ages 5–6) who are coming to terms with counting and the concept of days/time. Measurements of wing traits such as widths can be incorporated into ages 7–8 and for later years 9–11 incorporating measurements of wing areas and larval weights. High school students (13–18) can incorporate digital means to measure wingspan and colour and can create more complex and nuanced experimental designs. While at university, students can use this set-up for their own experiments to test across a range of temperatures and conditions. Here they can look at more advanced investigations of survival rates and colour change, and use geometric morphometrics to investigate shape characteristics to value-add to size measurements. Likewise, university students can follow up growth experiments with genetic investigations and behavioural assays to investigate underlying molecular mechanisms and behavioural consequences of the changes observed.

It is important to acknowledge some of the limitations of the set-up. The set-up in its current form is for investigating the impact of temperature only, with photoperiod kept constant between treatments (24 h). Temperature and photoperiod can interact to influence butterfly development [48]. Modifying to incorporate photoperiod could be done with a timer set at different day/night regimes, with the room temperature serving as the night-time temperature. The set-up fitted natural temperature ranges and predicted higher temperatures. If the investigation required cooler temperatures, then a cooling block could be used if the appropriate wattage is not available for the temperature range. Likewise, the application of the set-up in this study was conducted in a temperate country. Thus, if the experiments are conducted in classrooms without temperature control, then a cooling block maybe required. This would require some trial and error. For a school classroom, however, only two temperatures are required. 

Other cost savings can be made. Here we have used a full kit from a commercial company. Dry mix of painted lady food can be easily attainable in some countries. The dry mix can be bought in bulk, frozen and kept for up to a year when not in use, and made into an agar solution when required. The advantages of this set-up are that all of the other materials, including pots, utensils and pipettes, only need to be acquired once and can be reused. In particular, we would recommend investigating reusable glass vials for raising the larvae rather than using one-off disposable plastic containers. For a more natural approach, *V. cardui* has a broad host-plant range, with over 300 known species [49]. Many of the host plants are very common and typically belong to speciose and abundant plant families such as daisies/sunfowers/thistles (Asteraceae), nettles (Urticaceae) and hibiscus/mallows (Malvaceae). Guaranteeing that the amount of food is proportional for all larvae is a challenge with natural foods, but it makes for a more realistic experience. 

A final stage to working with the butterflies can be the release. Releasing the butterflies can be a powerful experience, as the students get the satisfaction of seeing the animals they have raised being set free. Given the cosmopolitan distribution of this species, the release of the butterflies is unlikely to cause major environmental damage. The opinion on the impact and value of the release of captive-bred species, even if they are common, is varied [50,51]. This is because there is little conclusive evidence to suggest a release of butterflies in general is harmful (although see: [52]), but equally, few data support a lack of impact [50,51]. Our broad recommendation is not to release the butterflies as the reasons for not releasing them are valid, including issues around genetic diversity, phenotypic robustness and disease transmission [53,54,55]. However, we do acknowledge the value of a release to the overall experience. Thus if a release is incorporated, then the following steps are best to adhere to: seek and follow local guidance, as some parts of the world do not support releases (e.g., Xerces Society policy in the US is against mass releases); only release males and then only individuals kept at room temperature, i.e., are not modified from normal size parameters; ask the sellers about the providence of the butterflies, i.e., have they been sourced from local areas and if not, then avoid releases; and lastly there is no need to release all individuals raised but instead a token number could be used. 

Lastly, the butterfly species chosen has a broad distribution and is highly migratory [25]. This means our incubator set-up with this species can be used in combination across the globe including across Europe, Asia, Africa and North America. It is seldom found in Australia, where a comparable species, the Australian painted lady, *V. kershawi*, could be used instead [45,48].

### 4.3. Active Learning in Environmental Education

Relative to this study, experiencing first-hand the effects of a physical environmental process (i.e., temperature and phenotypic shifts) in an active learning environment rather than using visual and auditory learning styles, will produce a more inherent understanding of the biological principle and thus gain a sense of urgency as it has been personally experienced. For example, a mixed-method study examined children aged 10 to 12 over a 15-week programme. The programme combined digital photography and hands-on educational activities focused on individual and collaborative change [56]. Children knew significantly more about the social and scientific dimensions of environmental change post programme and were motivated by their growing environmental impact awareness to take action to minimize environmental harm [56]. Similarly, related studies also identified that environmental engagement activities resulted in changes in skill, attitude and knowledge related to enhancing ecological, social and economic justice [57,58,59,60]. We feel this study provides the means for educators to give a real insight into the impacts of temperature on species survival. 

## 5. Conclusions

We showed the steps to design a homemade and cost-effective experiment for environmental education to show the effect of temperature on *V. cardui*. We tested the approach under experimental conditions and found the set-up to provide reliable and interesting findings. We suggest that to get the most out of the experiment findings, scenario-based activities should be incorporated, in which students need to brainstorm the ramifications of the findings in an ecological framework. The set-up is used for *V. cardui* but is adaptable for other study systems and could potential open avenues for environmental educators to test their own hypotheses. 

## Figures and Tables

**Figure 1 insects-15-00221-f001:**
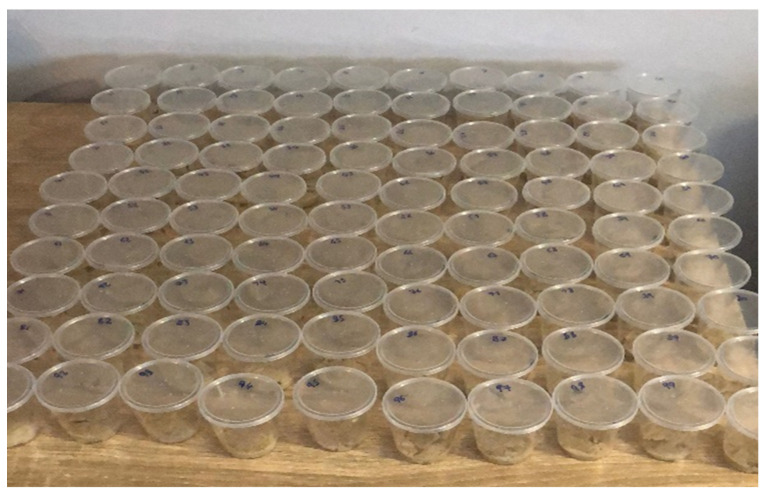
Image displaying all 100 *Vanessa cardui* larvae in individual vials.

**Figure 2 insects-15-00221-f002:**
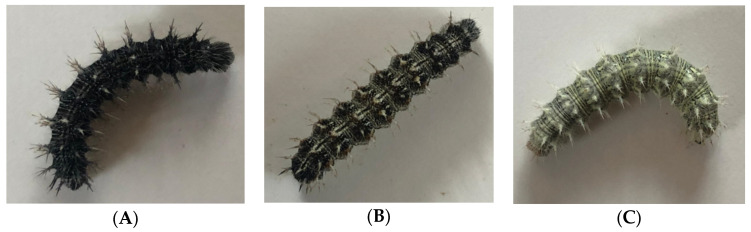
Colour code key used for phenological observations. Panel (**A**): Black (B); panel (**B**): Black/White (B/W); panel (**C**): White (W).

**Figure 3 insects-15-00221-f003:**
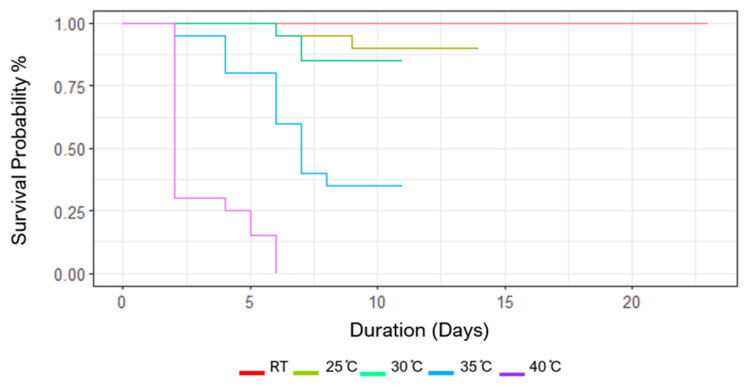
Survival probability of *Vanessa cardui* from fourth-instar larvae to emergence under differing temperature treatments (RT (room temperature, 18 °C), 25 °C, 30 °C, 35 °C, 40 °C).

**Figure 4 insects-15-00221-f004:**
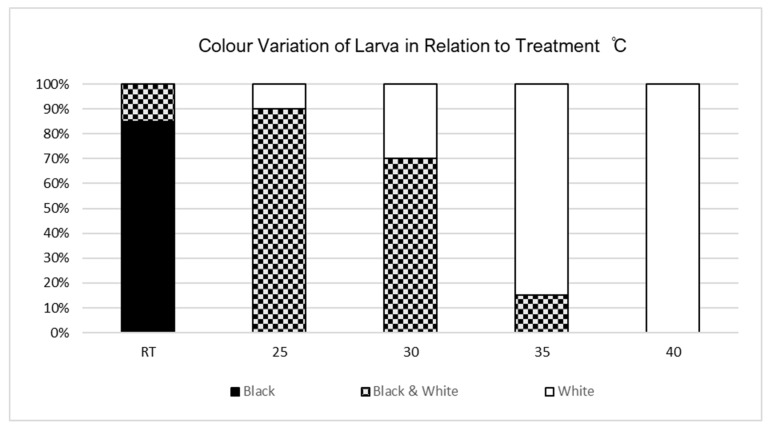
Colour variation of larvae in relation to temperature treatment.

**Figure 5 insects-15-00221-f005:**
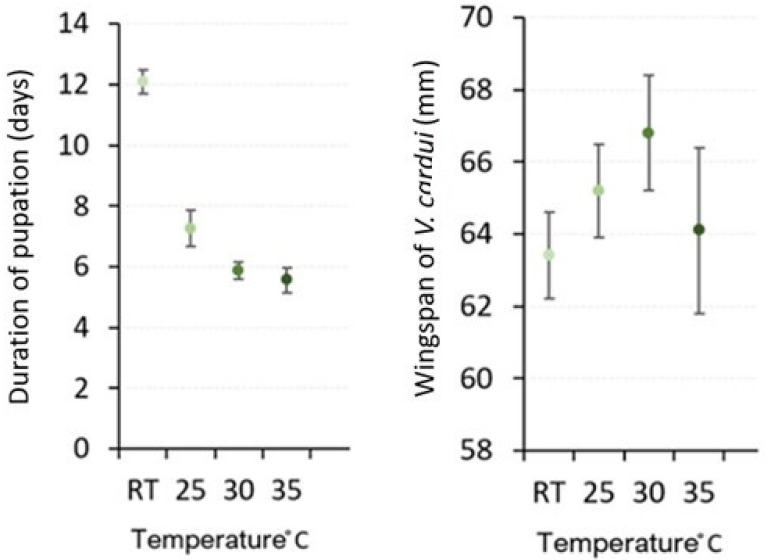
Pupation duration and wingspan in relation to temperature (means and SE are shown).

**Table 1 insects-15-00221-t001:** Per unit cost and source of materials for experimental incubator design, price relative to the time of purchase.

	Equipment	Site	Price (GBP) ***
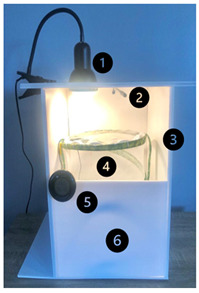	Basking clip-on spot lamp	Amazon	19.99
	Basking spotlight 50 W *	Reptilush	4.80
	HabiStat Dimming Thermostat	Swell UK	55.99
	20-pack A3 white Polystyrene foam boards	Amazon	18.99
	30 cm mesh habitat net **	Insect Lore	15.00
	Total		130.76

* for the 40 °C treatment, a 100 W spotlight was required which cost GBP 5.10; ** generally comes with the insect kit when purchasing the butterflies. Note the design used by Insect Lore is best for temporary use, and it is best to release the mature adults soon after maturity (pers. comm. M. Singer); *** These prices are current at the time of publishing and should be seen as a benchmark. It is best to source the materials locally where pricing may vary rather than relying on direct conversions of GBP to other currencies to determine costs. Numbers are explained in Section 2.2.

**Table 2 insects-15-00221-t002:** Results of the generalized linear mixed models with the response to pupation duration and wingspan (mm) of 80 *V. cardui*.

Response Variable	Predictor	Estimate	Std. Error	z-Value	*p*-Value
**Duration of pupation**	Intercept	2.492	0.013	190.88	<0.0001 ***
	25 °C	−0.509	0.021	−23.56	<0.0001 ***
	30 °C	−0.722	0.022	−32.68	<0.0001 ***
	35 °C	−0.778	0.033	−23.53	<0.0001 ***
**Wingspan of adults**	Intercept	4.148	0.007	548.4	<0.0001 ***
	25 °C	0.028	0.011	2.6	0.0095 **
	30 °C	0.525	0.121	4.3	0.0001 ***
	35 °C	0.012	0.016	0.7	0.465

** *p* < 0.01; *** *p* < 0.001.

## Data Availability

Raw data can be found in Appendix A.

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
