# Peer review of "Sensitivity of *Vanessa cardui* to Temperature Variations: A Cost-Effective Experiment for Environmental Education"

_insects, 2024, doi:10.3390/insects15040221_

Round 1

Reviewer 1 Report (Previous Reviewer 1)

Comments and Suggestions for Authors

The authors addressed all of my comments and I have no further suggestions to improve the paper. Thus, I think that the manuscript can be published “as it is”.

Author Response

Had no further suggestions and we would like to thank the reviewer for taking the time to oversee our manuscript again.

Reviewer 2 Report (New Reviewer)

Comments and Suggestions for Authors

I think this is a valuable study not just for classrooms but for shoestring budget entomologists trying to create their own incubators. The results also indicate interesting comparisons to monarch butterflies. However, I think there needs to be more citations and further detail added in the introduction.

Comments:

Painted ladies are very popular in the United States and Canada classrooms as well. I think you could reach a broader audience for your paper if you expand your focus outside of the UK. One suggestion would be to include the US dollar amount and Canadian dollar amount in the cost tables next to the £.

Line 92-93, 99-103, 110-111: citations needed

Line 218: extra space is needed between V. and cardui

L136-139: There are quite a few free versions of this food available. You could include a sentence mentioning alternative food sources since it may be less expensive for teachers to make their own rather than source from a breeder.

In the US schools have to meet education standards, usually the Next Generation Science Standards (NGSS). Is there an equivalent in the UK? If so, it might be helpful to include which standards this set-up could help meet. If your target audience for this paper is educators, they tend to search via standards (at least in the US) to locate lesson plans and experiment ideas. By including it, it could help them find your paper easier. Alternatively, maybe just including age ranges or grade levels that raising butterflies is common with could help reach the audience you want to meet. For example, in the US grades 2-3 (ages 7-8) are by far the most common range. If you include Canada as suggested earlier, you could also include their versions of state curriculum.

There are many sources stating migratory butterflies should not be released (see links listed below). Could you provide a more in-depth literature review on how as both a migrant and a cosmopolitan species that releasing V. cardui does not impact the overall health of the species? You also mention that this is a species that breeds intensively in a variety of habitats (L100-103) how does releasing species that do breed intensively not impact the local (wild) population structure of V. cardui? There is additional confusion as different butterfly rearing companies promote different practices. Carolina Biological Supply states “We do not advocate the release of organisms into the environment” compared to Nature Gift Store “you can release your butterflies outside after observing them”. I could not find a statement from Insect Lore. As such, providing guidance to teachers could help clear confusion. Alternatively, you could add a statement that suggests teachers do not released classroom reared insects.

https://images.peabody.yale.edu/lepsoc/nls/2010s/2010/2010_v52_n2.pdf#page=10

https://www.xerces.org/publications/policyposition-statements/monarch-butterflies-xerces-society-policy-on-butterfly

https://www.sciencedirect.com/science/article/abs/pii/S0006320714001566

https://royalsocietypublishing.org/doi/10.1098/rsbl.2019.0922

https://www.pnas.org/doi/10.1073/pnas.1904690116

https://royalsocietypublishing.org/doi/10.1098/rspb.2020.1326

https://royalsocietypublishing.org/doi/10.1098/rspb.2014.1734

L282-296: in this paragraph you state that wing morphology differences occur in migratory species dependent on species. Wouldn't this then be a negative outcome for these species to be released? If migratory phenotypes are being induced, then species should not be released.

Author Response

I think this is a valuable study not just for classrooms but for shoestring budget entomologists trying to create their own incubators. The results also indicate interesting comparisons to monarch butterflies. However, I think there needs to be more citations and further detail added in the introduction.

We would like to thank the reviewer for taking the time to oversee our manuscript and indicating its worthiness.

Comments:

Painted ladies are very popular in the United States and Canada classrooms as well. I think you could reach a broader audience for your paper if you expand your focus outside of the UK. One suggestion would be to include the US dollar amount and Canadian dollar amount in the cost tables next to the £.

Thank you for your suggestion. Given the butterflies are across the majority of continents except for Australia we decided to just go with the one currency as an exemplar. Costs will vary across the different countries, and it is likely to be cheaper or more costly in some countries. Hopefully cheaper as the UK cost of living is relatively high. However, to address the above suggestion about wider applicability we have made the following additions:

Lastly the butterfly species chosen has a broad distribution and is highly migratory (REF). This means our incubator set-up with this species can be used in combination across the globe including across Europe, Asia, Africa and North America. It is seldom found in Australia, where a comparable species, the Australian Painted Lady V. kershawi, could be used instead” 

We also discovered a factual error when writing this as previously the text indicated the species is found on all continents except Australia. Since that reference there has been incursions into Australia and the species is also largely absent from South America so we corrected this text.

And under the table for materials we wrote:

***Please note these prices are current at the time of publishing and should be seen as a benchmark. It is best to source the materials locally where pricing may vary rather than relying on direct conversions of pounds to other currencies to determine costs. 

Line 92-93, 99-103, 110-111: citations needed

DONE

Although we removed the release statement and just replaced it with  “Cosmopolitan and occurs naturally in large numbers

Line 218: extra space is needed between V. and cardui

DONE

L136-139: There are quite a few free versions of this food available. You could include a sentence mentioning alternative food sources since it may be less expensive for teachers to make their own rather than source from a breeder.

Based on this comment we provided the following paragraph. We did not recommend or provide a link to a product as the link is likely to come out of date pretty quickly and it is better to recommend general search terms.

Other cost savings can be made. Here we have used a full kit from a commercial company. Dry mix of Painted Lady food though can be easily attainable in some countries. The dry mix can be bought in bulk, frozen and kept for up to a year when not in use, and made into an agar solution when required. The advantages of this is that all of the other materials including pots, utensils and pipettes only need to be acquired once and can be reused. In particular we would recommend investigating reusable glass vials for raising the larvae rather than using one off plastic containers. For a more natural approach V. cardui has a broad hostplant range with over 300 species known (REF). Many of the hostplants are very common and typically belong to speciose and abundant plant families such as  Daisy/Sunfower/Thistles (Asteraceae), Nettle (Urticaceae) and Hibiscus/Mallow (Malvacease). Guarenteeing the amount of food is proportional for all larvae is a challenge with natural foods but it makes for a more realistic experience.

In the US schools have to meet education standards, usually the Next Generation Science Standards (NGSS). Is there an equivalent in the UK? If so, it might be helpful to include which standards this set-up could help meet. If your target audience for this paper is educators, they tend to search via standards (at least in the US) to locate lesson plans and experiment ideas. By including it, it could help them find your paper easier. Alternatively, maybe just including age ranges or grade levels that raising butterflies is common with could help reach the audience you want to meet. For example, in the US grades 2-3 (ages 7-8) are by far the most common range. If you include Canada as suggested earlier, you could also include their versions of state curriculum.

Thank you for the suggestion we chosen to make age recommendations rather than suggest education standards as these standards vary considerably between the countries that have painted ladies. Out next paper will illustrate the use in a class environment as this paper is introducing the tools.

There are many sources stating migratory butterflies should not be released (see links listed below). Could you provide a more in-depth literature review on how as both a migrant and a cosmopolitan species that releasing V. cardui does not impact the overall health of the species? You also mention that this is a species that breeds intensively in a variety of habitats (L100-103) how does releasing species that do breed intensively not impact the local (wild) population structure of V. cardui? There is additional confusion as different butterfly rearing companies promote different practices. Carolina Biological Supply states “We do not advocate the release of organisms into the environment” compared to Nature Gift Store “you can release your butterflies outside after observing them”. I could not find a statement from Insect Lore. As such, providing guidance to teachers could help clear confusion. Alternatively, you could add a statement that suggests teachers do not released classroom reared insects.

https://images.peabody.yale.edu/lepsoc/nls/2010s/2010/2010_v52_n2.pdf#page=10

https://www.xerces.org/publications/policyposition-statements/monarch-butterflies-xerces-society-policy-on-butterfly

https://www.sciencedirect.com/science/article/abs/pii/S0006320714001566

https://royalsocietypublishing.org/doi/10.1098/rsbl.2019.0922

https://www.pnas.org/doi/10.1073/pnas.1904690116

https://royalsocietypublishing.org/doi/10.1098/rspb.2020.1326

https://royalsocietypublishing.org/doi/10.1098/rspb.2014.1734

Thank you! The reviewer is absolutely right here we should have taken more care around the release aspect. And thank you for the direction to some useful papers here. Interestingly there are not really any papers that indicate V. cardui specifically is an issue and is possibly a nice student research study right there. But in general there are many concerns irrespective of the level of validation. WE chose to recommend not to release but had to acknowledge the value of it and if there was a release provided recommendations to consider.

“A final stage to working with the butterflies can be a release. Releasing the butterflies can be a powerful experience as the students get the satisfaction of seeing the animals they have raised being set free. Given the cosmopolitan distribution of this species the release of the butterflies is unlikely to cause major environmental damage. The opinion on the impact and value of the release of captive bred species, even if they are common, is varied (REF). This is because there is little conclusive data that suggests a release of butterflies in general is harmful (although see: REF) but equally there is little data supporting a lack of impact (REF). Our broad recommendation is not to release the butterflies as the reasons for not releasing them are valid such as issues around genetic diversity, phenotypic robustness and disease transmission (REF). We do acknowledge the value of a release to the overall experience though. So if a release is incorporated then the following is best to adhere too – seek and follow local guidance as the different parts of the world do not support releases (e.g. Xerces Society policy in the US is against mass releases); only release males and then only individuals kept at room temperature i.e. are not modified from normal size parameters; ask the sellers about the providence of the butterflies i.e. have they been sourced from local areas and if not avoid releases; and lastly there is no need to release all individuals raised but instead a token number could be used. 

Lastly the butterfly species chosen has a broad distribution and is highly migratory (REF). This means our incubator set-up with this species can be used in combination across the globe including across Europe, Asia, Africa and North America. It is seldom found in Australia, where a comparable species, the Australian Painted Lady, V. kershawi, could be used instead (REF).” 

L282-296: in this paragraph you state that wing morphology differences occur in migratory species dependent on species. Wouldn't this then be a negative outcome for these species to be released? If migratory phenotypes are being induced, then species should not be released.

Indeed and we have recommended only the release of individuals in the prevailing conditions ie room temperature if a release is incorporated at all.

This manuscript is a resubmission of an earlier submission. The following is a list of the peer review reports and author responses from that submission.

Round 1

Reviewer 1 Report

Comments and Suggestions for Authors

The reviewed study consists of two more or less independent parts. First, the authors designed and tested a simple cost-effective incubator that can be used for the experiments with thermal responses of various insects. Second, they conducted a number of very simple experiments on the thermal effects on different biological parameters of the painted lady butterfly, Vanessa cardui with particular emphasis on negative effects of extremely high temperatures in the context of the “global warming” problem. Generally speaking, both parts of the study can be interesting for certain groups of entomologists and therefore deserve to be published. However, the study has a number of serious flaws that should be addressed and fixed before publication (see below).

Major concerns

My most serious concern about the incubator is that the authors ignored such an important environmental factor as photoperiod (day length). Indeed, temperature was permanently controlled by turning on and off a spotlight. Thus, the insects were periodically lighted both during day and night time and, moreover, the period of lighting differed between experimental treatments (with the same lamp power, at higher temperatures the turn-on periods were longer). First, this very unnatural permanently flashing light can have a negative effect on any insect: the problem of “light pollution” is now considered as a very important biodiversity threat. Second, it is well known that photoperiod can have a significant specific effect on development of many insect species. In particular, this was demonstrated for the species congeneric to the model used in the present study [James, D. G. (1987). Effects of temperature and photoperiod on the development of Vanessa kershawi McCoy and Junonia villida Godart (Lepidoptera: Nymphalidae). Australian Journal of Entomology, 26(4), 289-292]. Thus, the authors should consider this problem and suggest some ways for its solution.

The second problem of their incubator is that it has no cooling block. In the present study, the used temperature gradient included “room temperature” that was 18.5°C. However, even in temperate climate zone (moreover in subtropical and tropical climate) this temperature can be easily maintained without a (rather expensive) conditioner only during winter but not during summer time. Thus, in reality the studies conducted with these incubators would include only high temperatures that would substantially distort the results: it is known that the effect of the global temperature increase on an insect can be negative near the southern limits of its geographic range but positive at the northern borders. I do understand that the cooling block would markedly increase the total cost of the incubator but I think that in any case the authors should consider this problem, too.

Regarding the “entomological part” of the study, the experiments were well conducted and the results look convincing but their novelty is low: similar thermal effects on various morphological parameters as well as on the rate of development were observed in many studies conducted on various insect species.

Minor corrections and comments

Line 18: There are two rather similar terms in the manuscript: “pupation time” and “pupation duration”. I guess that “pupation time” is the period from the beginning of the experiment to the pupal molt (i.e. to the pupa emergence) and “pupation duration” is the time of the next step: from the emergence of pupa to adult emergence. If this is true, “pupation duration” should be replaced by “the duration of the pupal stage” or something similar, to avoid confusion. And in any case the meaning of these terms should be clearly explained.

Line 79: Capitalize the first letter of a sentence.

Line 220: Delete the word “Displays”. The figure name should be “The survival probability of V. cardui from 4 th  instar larvae to emergence under different temperature treatments (RT-18 ̊C, 25 ̊C, 30 ̊C, 35 ̊C, 40 ̊C).”

Line 231: Delete the word “Displaying”. The figure name should be “Colour variation of larvae in relation to temperature.”

Lines 245-247: In addition to the GLM modeling (Table 2) the data on the duration of development should be treated with the standard way (linear regression of the rate of development on temperature) and the standard parameters (the lower developmental threshold and the sum of effective temperatures) should be calculated.

Line 249: Delete the word “Shows”. The figure name should be “Pupation duration and wingspan in relation to temperature (means and SE are shown)”.

Author Response

General comments

The reviewed study consists of two more or less independent parts. First, the authors designed and tested a simple cost-effective incubator that can be used for the experiments with thermal responses of various insects. Second, they conducted a number of very simple experiments on the thermal effects on different biological parameters of the painted lady butterfly, Vanessa cardui with particular emphasis on negative effects of extremely high temperatures in the context of the “global warming” problem. Generally speaking, both parts of the study can be interesting for certain groups of entomologists and therefore deserve to be published. However, the study has a number of serious flaws that should be addressed and fixed before publication (see below).

Major concerns

My most serious concern about the incubator is that the authors ignored such an important environmental factor as photoperiod (day length). Indeed, temperature was permanently controlled by turning on and off a spotlight. Thus, the insects were periodically lighted both during day and night time and, moreover, the period of lighting differed between experimental treatments (with the same lamp power, at higher temperatures the turn-on periods were longer). First, this very unnatural permanently flashing light can have a negative effect on any insect: the problem of “light pollution” is now considered as a very important biodiversity threat. Second, it is well known that photoperiod can have a significant specific effect on development of many insect species. In particular, this was demonstrated for the species congeneric to the model used in the present study [James, D. G. (1987). Effects of temperature and photoperiod on the development of Vanessa kershawi McCoy and Junonia villida Godart (Lepidoptera: Nymphalidae). Australian Journal of Entomology, 26(4), 289-292]. Thus, the authors should consider this problem and suggest some ways for its solution.

Thank you for that. Yes, that is a good point. And we have provided some text to address this and clarify some aspects --- “text modified The thermostat was responsible for keeping temperatures consistent within 0.9 ̊C for all temperature treatments, for example 25.0 – 25.9 ̊C. The thermostat automatically turns off the light briefly when temperatures are exceeded this threshold. However, the light was on consistently for 24hr and the thermostat was only triggered when the incubator was manipulated otherwise it remained stable when left alone.

The on and off was minimal as the temperature was varied mostly by the opening of the incubator. So, the on-off was fleeting and did not vary between treatments. The fleeting off is unlikely to impose any real tangible issue with development rate. Changes in developmental rate would be measured in differences of hours rather than fleeting periods. In terms of light pollution we take your point but light pollution influences activity rates primarily rather than any direct impact on development and indeed each individual ate the same amount of food within the same timeframe. Now saying all this we included some text to make sure this is more apparent

The second problem of their incubator is that it has no cooling block. In the present study, the used temperature gradient included “room temperature” that was 18.5°C. However, even in temperate climate zone (moreover in subtropical and tropical climate) this temperature can be easily maintained without a (rather expensive) conditioner only during winter but not during summer time. Thus, in reality the studies conducted with these incubators would include only high temperatures that would substantially distort the results: it is known that the effect of the global temperature increase on an insect can be negative near the southern limits of its geographic range but positive at the northern borders. I do understand that the cooling block would markedly increase the total cost of the incubator but I think that in any case the authors should consider this problem, too.

Yes agreed. There is a development threshold for these species and going lower will be just as detrimental as the higher temperature in a lack of development. The purpose of this study in reality was to garner an acceptable temperature range in which changes in phenology and morphology can be perceived without a big impact on survival. In the end this was so we owned the ethical issues and can thus recommend to teachers to use a more refined temperature range. We added a new paragraph: “It is important though to acknowledge some of the limitations of the set-up. The set-up in its current form is for investigating the impact of temperature only with photoperiod kept constant between treatments (24hr). Temperature and photoperiod can interact to influence butterfly development [46]. Modifying to incorporate photoperiod could be done with a timer set at different day-night regimes with the room temperature serving as the nighttime temperature. The set-up fitted natural temprature ranges and predicted higher temperatures. If the investigation required cooler temperatures then a cooling block could be used if the appropriate wattage is not available for the temperature range. Likewise the application of the set-up in this study was conducted in a temperate country. Thus if the experiments are conducted in classrooms without temperature control then a cooling block maybe required. This would require some trial and error. For a school classroom though only two temperatures are required.”   

Regarding the “entomological part” of the study, the experiments were well conducted and the results look convincing but their novelty is low: similar thermal effects on various morphological parameters as well as on the rate of development were observed in many studies conducted on various insect species.

Thank you for that and the comment about novelty is fair. This is in a sense due to the scope of this study which was to create an approach that educators could adopt. Although we are happy to accept the novelty limitation it is worth noting though that the findings regarding size and duration rates is not equivocal in insects or even in butterflies. And further to this it provides more evidence to challenge the generality of the temperature-size rule. This rule continues to be rolled out as a done deal and for vertebrates this maybe the case with large versions of species being found in cold areas and smaller versions in warm areas. But there is growing evidence with this paper as an example in which the outcome is a reversal of the prediction of the temperature-size rule.

Editing

Line 18: There are two rather similar terms in the manuscript: “pupation time” and “pupation duration”. I guess that “pupation time” is the period from the beginning of the experiment to the pupal molt (i.e. to the pupa emergence) and “pupation duration” is the time of the next step: from the emergence of pupa to adult emergence. If this is true, “pupation duration” should be replaced by “the duration of the pupal stage” or something similar, to avoid confusion. And in any case the meaning of these terms should be clearly explained.

Thank you for picking that up – it seems we rarely used the term pupation time in the document except for the abstract but used pupation duration a lot. With respect to where pupation was used we felt it was better to place development time as we rarely indicated the rate to pupation as that was broken down into instar development times but instead pupation was used more in the context of the length of pupation = pupation duration.

Line 79: Capitalize the first letter of a sentence.

Took a little to find as it was 81 on my document hehe – but done!

Line 220: Delete the word “Displays”. The figure name should be “The survival probability of V. cardui from 4 th  instar larvae to emergence under different temperature treatments (RT-18 ̊C, 25 ̊C, 30 ̊C, 35 ̊C, 40 ̊C).”

Done

Line 231: Delete the word “Displaying”. The figure name should be “Colour variation of larvae in relation to temperature.”

Done

Lines 245-247: In addition to the GLM modeling (Table 2) the data on the duration of development should be treated with the standard way (linear regression of the rate of development on temperature) and the standard parameters (the lower developmental threshold and the sum of effective temperatures) should be calculated.

Regarding this point, we would like to point out that a linear regression would not be the best approach to show the relationship between development and temperature since 1) we have four controlled temperatures and not a continuum; 2) it is clear from figure 5 that the relationship is not linear.

Line 249: Delete the word “Shows”. The figure name should be “Pupation duration and wingspan in relation to temperature (means and SE are shown)”.

Done

Reviewer 2 Report

Comments and Suggestions for Authors

I have often considered pouring through research on arthropods to identify possible demonstrations and experiments that could be replicated in instructional environments. The use of insects and other arthropods for instruction in and outside of formal schooling is effective at several different levels. First, there are few ethical issues and insects are everywhere. Ease of use is coupled with immediate informal/unplanned experiences outside the classroom, something not possible with lessons about penguins or polar bears. There is some evidence that arthropods (bugs in the vernacular) are widely disliked (fear/disgust) and these negative reactions limit outdoor activity. When students are given opportunities to study insects, their aversion to bugs declines. This study uses butterflies which are what I call positively experienced story-book bugs (butterflies, lady bugs, snails, crickets, etc), but the general public is far less comfortable with the larval stages of butterflies. Arthropods have short life-cycles making all sorts of demonstrations and experiments possible in the short-span of classroom instruction.  

My point above is that there are multiple layers of justification for this type of instruction beyond illustrating the effects of climate change.  

This paper provides in detail what is essentially a lesson plan not in lesson plan format. All the components of a lesson plan are present. There is somewhat vague discussion of grade/age level differences in instruction. This experiment requires a number of ancillary skills (measurement, math, lab skills) that would restrict what level of student could accomplish the instructional objectives. I would encourage some attention to this issue.  

There does not seem to be an actual application of this activity in the classroom with any sort of systematic evaluation, or even a critique from classroom teachers about its usefulness or practicality. This needs to be acknowledged and the comments about it being a “powerful” activity revised to being a potentially powerful activity.

Comments on the Quality of English Language

The paper is well written for scientists. To make this useful to teachers would require a format and writing level more akin to what would be found in "Science Teacher" magazine or converted to a lesson plan format typical of what is provided teachers.

Author Response

I have often considered pouring through research on arthropods to identify possible demonstrations and experiments that could be replicated in instructional environments. The use of insects and other arthropods for instruction in and outside of formal schooling is effective at several different levels. First, there are few ethical issues and insects are everywhere. Ease of use is coupled with immediate informal/unplanned experiences outside the classroom, something not possible with lessons about penguins or polar bears. There is some evidence that arthropods (bugs in the vernacular) are widely disliked (fear/disgust) and these negative reactions limit outdoor activity. When students are given opportunities to study insects, their aversion to bugs declines. This study uses butterflies which are what I call positively experienced story-book bugs (butterflies, lady bugs, snails, crickets, etc), but the general public is far less comfortable with the larval stages of butterflies. Arthropods have short life-cycles making all sorts of demonstrations and experiments possible in the short-span of classroom instruction.  

Agreed – and if we could do the same experiment with something other than butterflies I would certainly have embraced it but it is more about what a school can manage and what the experience is for the children.

My point above is that there are multiple layers of justification for this type of instruction beyond illustrating the effects of climate change.  

Agreed. Our real interest is experiential learning but here we wanted to tackle a concept that is not as easy to access for the general public which is phenological shifts. Things living or dying is an easier concept to consider than the more nuanced aspects of species surviving but with sub-optimal opportunities to eat and breed leading to insidious slow declines.

This paper provides in detail what is essentially a lesson plan not in lesson plan format. All the components of a lesson plan are present. There is somewhat vague discussion of grade/age level differences in instruction. This experiment requires a number of ancillary skills (measurement, math, lab skills) that would restrict what level of student could accomplish the instructional objectives. I would encourage some attention to this issue.  

There does not seem to be an actual application of this activity in the classroom with any sort of systematic evaluation, or even a critique from classroom teachers about its usefulness or practicality. This needs to be acknowledged and the comments about it being a “powerful” activity revised to being a potentially powerful activity.

Thank you for that. What you suggest is the subject of the next paper on work that has also been conducted. This paper is to outline the novel set-up and provide the pilot study for its use to address a research question but also to indicate the suitable temperature ranges for investigating the impacts of temperature and development. For most cases the students would only use two temperatures in which the rate of change is obvious and there is a minimal of loss of life. And indeed, the use of the activity in the classroom primary school students just used two temperatures. To cover the set-up and the pilot study was too much to present with the school trial as the school trial was also about how the experience led to attitude changes.

Comments on the Quality of English Language

The paper is well written for scientists. To make this useful to teachers would require a format and writing level more akin to what would be found in "Science Teacher" magazine or converted to a lesson plan format typical of what is provided teachers.

Point taken. The paper though is a scientific investigation using tools that a teacher could implement for their own design. The study is investigating the rigour of this design with the mindset of using the approach for classroom activities and for that reason the language needs to be scientific. We are not school educators and thus it is not for us to create lesson plans namely because curricula vary between different countries and this study is not meant for any one education system. Likewise, the approach taken here can be scaled down or up depending on the age group and at what level certain foundation skills are delivered.  As indicated above the next publication will provide greater guidance as it is on the application of the study in a school environment and the outcomes of that.